# Crohn’s Disease Increases the Mesothelial Properties of Adipocyte Progenitors in the Creeping Fat

**DOI:** 10.3390/ijms22084292

**Published:** 2021-04-20

**Authors:** Ana Madeira, Carolina Serena, Miriam Ejarque, Elsa Maymó-Masip, Monica Millan, M. Carmen Navarro-Ruiz, Rocío Guzmán-Ruiz, María M. Malagón, Eloy Espin, Marc Martí, Margarita Menacho, Ana Megía, Joan Vendrell, Sonia Fernández-Veledo

**Affiliations:** 1Department of Research, Institut d’Investigació Sanitària Pere Virgili, Hospital Universitari de Tarragona Joan XXIII, 43005 Tarragona, Spain; madeira.aps@gmail.com (A.M.); carolserena@gmail.com (C.S.); miriam.ejarque@gmail.com (M.E.); elsamaymomasip@gmail.com (E.M.-M.); monica.millan@ymail.com (M.M.); amegia.hj23.ics@gencat.cat (A.M.); 2Instituto de Salud Carlos III, CIBER de Diabetes y Enfermedades Metabólicas Asociadas (CIBERDEM), 08029 Madrid, Spain; 3Department of Surgery, Colorectal Surgery Unit, Hospital Universitari i Politècnic La Fe, 46026 Valencia, Spain; 4Instituto Maimónides de Investigación Biomédica de Córdoba (IMIBIC), University of Cordoba, Reina Sofia University Hospital, Edificio IMIBIC, Avda. Menéndez Pidal s/n, 14004 Cordoba, Spain; b02narum@uco.es (M.C.N.-R.); bc2gurur@uco.es (R.G.-R.); bc1mapom@uco.es (M.M.M.); 5Department of Cell Biology, Physiology and Immunology, University of Cordoba, 14004 Cordoba, Spain; 6CIBER Fisiopatología de la Obesidad y Nutrición (CIBEROBN), Instituto de Salud Carlos III, 28029 Madrid, Spain; 7Colorectal Surgery Unit, General Surgery Service, Hospital Vall d’Hebron, Universitat Autònoma de Barcelona, 08035 Barcelona, Spain; eespin@vhebron.net (E.E.); marcmartig@gmail.com (M.M.); 8Digestive Unit, Hospital Universitari Joan XXIII, 43007 Tarragona, Spain; mmenacho.hj23.ics@gencat.cat; 9Department Medicine and Surgery, Medicine School, Universitat Rovira i Virgili, 43003 Tarragona, Spain

**Keywords:** Crohn’s disease, adipose tissue, mesothelium, adipose-derived stem cells

## Abstract

Our understanding of the interplay between human adipose tissue and the immune system is limited. The mesothelium, an immunologically active structure, emerged as a source of visceral adipose tissue. After investigating the mesothelial properties of human visceral and subcutaneous adipose tissue and their progenitors, we explored whether the dysfunctional obese and Crohn’s disease environments influence the mesothelial/mesenchymal properties of their adipocyte precursors, as well as their ability to mount an immune response. Using a tandem transcriptomic/proteomic approach, we evaluated the mesothelial and mesenchymal expression profiles in adipose tissue, both in subjects covering a wide range of body-mass indexes and in Crohn’s disease patients. We also isolated adipose tissue precursors (adipose-derived stem cells, ASCs) to assess their mesothelial/mesenchymal properties, as well as their antigen-presenting features. Human visceral tissue presented a mesothelial phenotype not detected in the subcutaneous fat. Only ASCs from mesenteric adipose tissue, named creeping fat, had a significantly higher expression of the hallmark mesothelial genes mesothelin (*MSLN*) and Wilms’ tumor suppressor gene 1 (*WT1*), supporting a mesothelial nature of these cells. Both lean and Crohn’s disease visceral ASCs expressed equivalent surface percentages of the antigen-presenting molecules human leucocyte antigen—DR isotype (HLA-DR) and CD86. However, lean-derived ASCs were predominantly HLA-DR ^dim^, whereas in Crohn’s disease, the HLA-DR ^bright^ subpopulation was increased 3.2-fold. Importantly, the mesothelial-enriched Crohn’s disease precursors activated CD4^+^ T-lymphocytes. Our study evidences a mesothelial signature in the creeping fat of Crohn’s disease patients and its progenitor cells, the latter being able to present antigens and orchestrate an immune response.

## 1. Introduction

Excessive and dysfunctional adipose tissue is responsible for many of today’s health epidemics, especially obesity. Additionally, obesity-associated chronic low-grade inflammation is a key contributing factor to systemic metabolic dysfunction and comorbidities such as type 2 diabetes (T2D), fatty liver disease, cardiovascular disease, and cancer [1]. Fat accumulation and inflammation can also be locally restricted. Such is the case in Crohn’s disease, a chronic inflammatory bowel disease characterized by the expansion of mesenteric adipose tissue, known as “creeping fat”, which correlates with the disease severity [2,3,4]. Recent findings indicate similarities between the complex pathological features of Crohn’s disease and obesity/T2D with respect to adipose tissue behavior [5], especially in terms of inflammation [5,6,7], and it is clear that adipose tissue holds yet unrealized functions in the setting of these diseases. In fact, a recent publication used a novel mesenteric creeping fat index based on computed tomography to characterize accurately the extent of mesenteric fat wrapping in surgical specimens, which correlated with intestinal fibrostenosis [8].

White adipose tissue (WAT) is a highly plastic and dynamic tissue composed of various cell types, including mature adipocytes, adipose-derived stem cells (ASCs, the precursor cells that give rise to adipocytes), vascular cells, and a diverse array of innate and adaptive immune cells. In addition to being a highly active endocrine organ that can orchestrate whole-body energy homeostasis [9,10], WAT participates in immune surveillance and host defense. Indeed, recent research placed mammalian WAT at the center of immunological memory: WAT is a reservoir of memory T-cells, which are able to sustain long-term immune responses to infection [11].

Increasing evidence suggests that an appreciation of the complex heterogeneity among adipose tissue progenitors is key to fully understand the functional and biological differences between fat depots [12,13,14,15,16,17]. Depending on its anatomical location, WAT can be broadly classified as visceral adipose tissue (VAT) or subcutaneous adipose tissue (SAT). Global differences in the gene expression and functional properties (including metabolic activity and adipokine secretion profile) of adipose tissue are well-documented for visceral and subcutaneous depots [18]. Nevertheless, significant gaps remain in our understanding of the developmental origins of the different WAT depots, including creeping fat in Crohn’s disease patients. Recent evidence indicates that creeping fat expands to prevent the dissemination of translocated gut bacteria across the leaky intestinal barrier [19].

Adipocytes are generally believed to derive from the mesoderm, and distinct developmental timings are described for SAT and VAT [9,20]. Additionally, adipocytes have long been thought to derive exclusively from tissue-resident ASCs/mesenchymal stem cells (MSCs); however, there is growing evidence that individual adipose tissue depots may arise through distinct lineages [9,13,14,15,16,21,22,23,24,25]. The mesothelium—a layer of specialized epithelial (mesothelial) cells that forms the lining of all serous cavities—was recently proposed as a source of adipocyte progenitors for VAT, but not SAT, hinting at the diversity among visceral adipocytes and precursor cells [17,21]. A recent single-cell sequencing study identified a mesothelial subpopulation within the human visceral stromal vascular fraction [17]. However, little is known about the functionality of these mesothelial-enriched precursors, and whether their proportion is changed by a pathological environment. Mesothelial cells are capable of recognizing pathogens and tissue damage and initiating inflammatory responses through antigen presentation and cytokine production [26,27,28,29,30,31]. These cells are thought to represent the first line of defense against bacterial insults to the adipose tissue [32], which is a particularly troublesome event in Crohn’s Disease and is possibly the trigger of creeping fat expansion [19].

The mesothelium is one of the fundamental histological structures that composes the mesentery [33]. Mesentery´s fat is abnormally expanded in Crohn’s Disease, leading to the formation of creeping fat around the intestinal damaged area. Given the recent link established between the mesothelium and the development of VAT, we hypothesized that human visceral ASCs, especially those from the creeping fat, derive from the mesothelial lineage. Upon microbial dysregulation and the breakdown of the intestinal barrier (both hallmarks of Crohn’s Disease), ASCs migrate towards the intestinal lesions to participate in tissue repair [19]. This event, which would be beneficial in the first instance, also induces an accumulation of adipose tissue surrounding the intestine (“creeping fat”), which is a consequence of the natural fate of ASCs to differentiate into adipocytes. In this work we hypothesized that the proportion of mesothelial-derived precursors is increased in the creeping fat and that the newly recruited precursors are able to process and present antigens and mount an immune response. Accordingly, we set out to investigate the mesothelial/mesenchymal properties of human adipose tissue and their precursors in Crohn’s disease and obesity (an established inflammatory condition). Finally, our goal was to demonstrate that ASCs’ populations with different abundances of mesothelial precursors would also differ in their immune phenotypic and antigen-presentation features.

## 2. Results

### 2.1. A Mesothelial Gene Signature Distinguishes Visceral Adipose Tissue and Its Precursors from Subcutaneous Adipose Tissue

To investigate the mesothelial properties of human adipose tissue, we first surveyed the expression of a panel of mesothelial marker genes in the SAT and VAT depots of lean individuals using an inventoried TaqMan array. We found that the expression of keratin 8 *(KRT8*), cadherin 1 (*CDH1*), mesothelin (*MSLN*), leucine rich repeat neuronal 4 (*LRRN4*), and Wilms’ tumor suppressor gene 1 (*WT1*) was significantly higher in VAT than in SAT (Figure 1A), which is consistent with the finding of a mesothelial gene expression pattern in omental fat in a previous microarray-based analysis (GEO accession number GSE20950; Appendix A) [34]. By contrast, no differences were found in the expression of typical mesenchymal genes between VAT and SAT samples (Figure 1A).

We then performed qualitative liquid chromatography coupled with tandem mass spectrometry (LC-MS/MS) proteomic analysis in lean adipose tissue to gain further insight into the differences in mesothelial marker expression between SAT and VAT. Results showed that the mesothelial proteins MSLN, tight junction protein 1 (TJP1), keratin 7 (KRT7), KRT8, and calponin 1 (CNN1) were detected in VAT but not in SAT samples (Appendix A), in accordance with the transcription profile.

Having established a mesothelial signature for VAT from lean subjects, we extended this analysis to examine the gene expression profile of precursor cells isolated from the two distinct adipose tissue depots. As shown in Figure 1B, expression of the mesothelial genes *CNN1*, *CDH1*, *MSLN*, and *WT1* was significantly higher in ASCs obtained from lean intra-abdominal adipose tissue than from SAT, supporting a mesothelial nature for these cells. Moreover, a reduction in the expression of mesenchymal markers was also noted for lean visceral-derived ASCs when compared with their SAT counterparts [17,35,36,37]. Overall, these results support a mesothelial origin for human VAT and its precursors.

### 2.2. Creeping Fat Precursor Cells Have Mesothelial-Enriched Properties

Obesity and Crohn’s disease are distinguished by intra-abdominal fat accumulation, which is a well-recognized forerunner of disease pathology [5]. Given our identification of a mesothelial profile for VAT in lean subjects, we next tested whether this held true in the context of obesity and Crohn’s Disease. We again analyzed a panel of marker genes in adipose tissue and ASCs obtained from obese and morbid obese subjects, and from patients with Crohn’s Disease. We found a consistent upregulation of multiple mesothelial markers in VAT samples when compared with equivalent SAT samples, and these differences were maintained irrespective of the BMI or whether samples were from patients with Crohn’s disease (Figure 2 and Figure 3). We also observed a non-significant downregulation in the expression of several mesenchymal genes, when comparing VAT and SAT samples from obese, morbid obese, and Crohn’s disease patients. No significant differences were observed in the expression of mesothelial markers in obese, morbid obese, and Crohn’s disease VAT samples with respect to lean-derived VAT (Appendix A). Interestingly, the expression of several mesothelial genes in mesenteric fat-derived ASCs from patients with Crohn’s disease showed a clear upregulation with respect to equivalent lean VAT-derived ASCs, namely *LRRN4*, carbonic anhydrase 2 (*CA2*), *MSLN*, and *KRT8* (Figure 4), with a trend for greater expression for others including *CDH1*, *KTR7*, *WT1*, and *THBD*. No differences were observed for the expression of mesenchymal markers in obese, morbid obese, and Crohn’s disease VAT and VAT-derived ASCs with respect to equivalent lean-derived samples (Appendix A).

Our results thus suggest that the abnormal inflammatory environment in creeping fat enriches the mesothelial precursor subpopulation.

### 2.3. Creeping Fat-Derived ASCs Have an Altered Immunophenotypic Profile and Act as Antigen-Presenting Cells

Having identified a mesothelial origin for VAT-derived ASCs, especially for ASCs from creeping fat, we sought to investigate the ability of these cells to mount an immune response. We started by employing flow cytometry to evaluate the expression profile of cell-surface antigens in both SAT- and VAT-derived ASCs of lean individuals and define their immune phenotype. Overall, SAT- and VAT-derived ASC populations equally expressed a common set of mesenchymal surface antigens, namely CD73, CD90, CD105, and CD36, and presented negligible expression of hematopoietic and endothelial markers (CD31, CD34, CD14, and CD45), complying to the International Society of Cell Therapy criteria (Appendix A). Nevertheless, the percentage of surface expression and the mean fluorescence intensity (MFI) of the antigen-presenting molecules human leucocyte antigen—DR isotype (HLA-DR) and CD86 were markedly increased in VAT-derived ASCs compared to SAT-derived precursors from lean subjects (Figure 5A).

Previously, we demonstrated that Crohn’s disease-derived ASCs had impaired immunomodulatory capacity, which was linked to the creeping fat-induced inflammatory milieu [7]. In this work, we investigated if lean and Crohn’s disease VAT-derived ASCs, bearing differential mesothelial properties could be endowed with distinct antigen presentation features. Interestingly, both lean and Crohn’s disease VAT-derived ASCs expressed equivalent surface percentages of antigen-presenting molecules HLA-DR and CD86, and presented similar MFI values for these markers (Figure 5B). Yet, remarkably, the surface expression of CD40, a co-stimulatory protein required for antigen presentation [38], was mainly detectable in Crohn’s disease-derived ASCs. Moreover, quantitative flow cytometric analysis revealed two distinct subpopulations of HLA-DR (HLA-DR ^dim^/HLA-DR ^bright^) that were differently distributed within lean and Crohn’s disease ASCs’ populations (Figure 5C). Lean-derived ASCs were predominantly HLA-DR ^dim^, whereas in Crohn’s disease ASCs, there was a 3.2-fold increase in the HLA-DR ^bright^ subpopulation with respect to their lean counterparts, although not statistically significant. Despite presenting similar mesenchymal, hematopoietic, and endothelial expression profiles (Appendix A), visceral lean- and Crohn’s disease-derived ASCs immune phenotypic differences hint for divergent immunoregulatory properties. Therefore, we next explored lean and Crohn’s disease-derived ASCs immunomodulatory properties by investigating their effect on CD4^+^ T-lymphocytes proliferation, as measured by CellTrace violet dye (CTV) intensity after two days of co-culture with autologous ASCs (Figure 6A). We compared the effect of Crohn’s disease-derived ASCs from three donors on autologous CD4^+^ T-lymphocytes proliferation to the effect produced by a healthy subject’s precursor cells. CD4^+^ T-lymphocytes’ proliferation was measured in co-culture with (i) untreated ASCs, (ii) ASCs pulsed with the antigen ovalbumin (OVA) and (iii) ASCs primed with Interferon gamma (IFNγ). Healthy-derived ASCs were unable to activate CD4^+^ T-lymphocytes, whereas Crohn’s disease-derived ASCs increased CD4^+^ T-cells proliferation by approximately 3%, which was associated with an increase in IL-2 (3.325 pg/mL vs. 9.615 pg/mL, respectively), a cytokine released by activated CD4^+^ T-cells. These results are in agreement with CD4^+^ T-cells’ activation by bone marrow-derived mesenchymal stem cells (Figure 6B–D) [39,40]. Pulsing Crohn’s disease-derived ASCs with OVA prior to the co-culture further stimulated T-cells’ proliferation, whereas priming with IFNγ had no effect on T-cells’ activation. Neither OVA nor IFNγ exerted any effect on lean-derived ASCs.

It is described that MSCs promote T-cells’ survival by exerting an anti-apoptotic effect [41]. Accordingly, we observed that healthy-derived ASCs increased T-cells’ viability (Figure 6E), an effect that was practically abolished by stimulation with OVA and IFNγ. Interestingly, Crohn’s disease ASCs induced a marked reduction on lymphocytes’ viability, which was maintained with OVA and IFNγ treatments, and is in accordance with the link between proliferation and apoptosis described for activated lymphocytes [42].

## 3. Discussion

To our knowledge, this is the first study to establish a mesothelial profile in creeping fat and its precursors, suggesting a mesothelial origin of this tissue. Our study supports previous data pointing towards the mesothelial signature of visceral fat [17,21], and further identifies the mesothelial-enriched fat precursors as key drivers of creeping fat in Crohn’s disease. Remarkably, mesothelial-enriched creeping fat-derived ASCs performed as unconventional antigen-presenting cells and activated CD4^+^ T-lymphocytes. Thus, our results suggest that the unique environment of mesenteric fat in Crohn’s disease favors the recruitment/proliferation of mesothelial progenitor cells, which underscores creeping fat’s unappreciated role in orchestrating adaptive immune responses.

Adipose tissue has a remarkable capacity to expand or remodel in response to physiological cues [43]. The complexity of WAT was only recently appreciated because of our increased understanding of the distinct functions of individual WAT depots and the differences between adipocytes within depots [9]. Indeed, recent research indicates that individual depots contain adipocytes that can arise through distinct lineages [14,43]. In line with this notion, our tandem transcriptomic/proteomic approach revealed a mesothelial provenance for human VAT, which is in agreement with growing evidence in the literature: (i) publically available microarray data for human adipose tissue describing mesothelial gene expression (microarray—GEO accession number GSE37324—Appendix A) [34,35], (ii) differentially-methylated loci for CpG sites located within mesothelial markers (*WT1*, *MSLN*, and Uroplakin 3B) between subcutaneous and visceral adipose tissue [44], (iii) lineage-tracing studies in rodents [21], and (iv) single-cell analysis of the human stromal vascular fraction [17].

The precise embryonic origin of human WAT remains unclear, and our study attempts to enrich our understanding of the heterogeneous nature of human VAT. Whether the mesothelial properties of human VAT are embryonic in origin or arise from the differentiation of neighboring mesothelial cells, or both, remains to be elucidated. It is believed that the precursor populations involved in the initial development of adipose tissue (“organogenesis”) are distinct from precursor populations in adult animals tasked with the maintenance and expansion of adipose depots [45], and mesothelial cells are no exception. In addition to those studies tracing rodent VAT from embryogenic mesothelial precursors [21,46], adult mesothelial cells isolated from human and adult rodents are known to differentiate along the vascular smooth muscle cell lineage and undergo chondrogenic, osteogenic, and, importantly, adipogenic differentiation [47,48,49,50,51]. This suggests that adult mesothelial cells retain embryonic multipotency and could represent a population of primitive mesodermal stem cells [50,52].

The origin of fat cells is a fundamental biological question with important ramifications for human health and disease, not only in the context of the global obesity epidemic, but also in understanding the origin of hyperplasic mesenteric fat in Crohn’s disease. Having established that visceral ASCs are a heterogeneous population with both mesenchymal and mesothelial properties, we aimed to investigate whether these precursors could be differentially activated according to the pathological cues prevailing in dysfunctional adipose tissue, as in obesity and Crohn’s disease. Our findings show that despite the homogeneity of the mesenchymal [36,37] and mesothelial gene signatures in the VAT of lean, obese, and morbid obese subjects and patients with Crohn’s Disease, the mesothelial profile of ASCs derived from creeping mesenteric fat is markedly different from their non-obese counterparts. The significant upregulation of mesothelial markers in mesenteric fat-derived ASCs points to the recruitment of progenitors from the adipose-lining mesothelium, which could contribute to the unique properties of creeping fat; the unaltered mesothelial traits among obese-derived ASCs compared to lean indicate that adipose tissue pathologies impact differently on their precursor cells.

We recently demonstrated that creeping fat-derived ASCs have unique immune and biological traits when compared with equivalent cells from healthy tissue, including a higher phagocytic capacity [7]. The major source of antigen-presenting cells in adipose tissue remains controversial [53,54,55], and there is no consensus as to whether ASCs present antigens [56,57]. Remarkably, mesothelial cells are able to phagocytose, present antigens, and induce T-cell proliferation [26,27,28,29,30,31]. Therefore, we speculate that the heterogeneity among adipose tissue precursors and depots could account for the discrepancies regarding the antigen-presenting potential of ASCs. Most remarkably, we observed that mesothelial-enriched precursors from creeping fat stimulated CD4^+^ T-lymphocytes’ proliferation, functioning much like antigen-presenting cells (APCs). This is in direct contrast with ASCs’ generally appraised ability in modulating the immune system by combining potent immunosuppressive properties with low immunogenicity. It is widely accepted that unstimulated ASCs do not express major histocompatibility complex (MHC) class II molecules (such as HLA-DR) [58,59], which are mainly present on the surface of professional APCs and are required to present exogenous antigens to CD4^+^ T-lymphocytes in conjunction with co-stimulatory signals (such as CD86 and CD40) [60]. MSCs can only adopt an APC-like function if exposed to low levels of IFNγ that upregulates MHC-II [40], whereas IFNγ at high concentrations downregulates MHC-II expression and licenses MSCs’ immunosuppressive functions [61]. However, we demonstrate that creeping fat-derived ASCs, even without stimulation, are endowed with all the molecular elements of the MHC class II machinery which ultimately compromises their immunosuppressive capacity. Surprisingly, we detected moderate levels of HLA-DR in lean SAT-derived ASCs, and high levels of both HLA-DR and CD86 in VAT-derived precursor cells from lean subjects. Yet, given that lean-derived ASCs are predominantly HLA-DR ^dim^ with negligible expression of antigen presentation of co-stimulatory molecules, such as CD40, these cells are ultimately unable to activate CD4^+^ T-lymphocytes. Interestingly, healthy-derived ASCs increased CD4^+^ T-cells’ viability, which is in accordance with MSCs protecting T-cells from apoptosis, maintaining them in a state of quiescence [41]. Although Crohn’s disease-derived ASCs increased T-cells’ proliferation, they had an inverse effect on their viability, possibly as part of a regulatory mechanism of T-cell expansion. Once activated, T-cells go through one, and perhaps several, cell cycles and enter a late G1 or S phase, after which they become exquisitely susceptible to apoptosis [42]. This ensures an effective defense while avoiding an uncontrolled immune response.

We acknowledge that our observations regarding the antigen-presenting properties of Crohn’s disease-derived ASCs would benefit from a larger sample size. This work had the daunting task of coordinating the collection/culturing of visceral ASCs (obtained through invasive surgical procedures) with the availability of blood samples for autologous CD4^+^ T-lymphocytes’ isolation. Importantly, this represented a challenge at the time of gathering proper healthy/control samples, especially given the preliminary evidences that inflammation (measured by c-reactive protein), body-mass index, and glucose tolerance seem also to correlate with the ability of ASCs to stimulate adaptive immune responses (unpublished data). The variability/heterogeneity between samples could add another layer of complexity when interpreting results. Factors such as disease activity, extent, location, behavior, and therapies could also be relevant to understand fully the interaction between precursor cells and immunity. Another potential limitation of our study is that trace amounts of mesothelium might have remained during the processing of VAT samples to discard the mesentery, specifically. Although this could arguably account for the up-regulated mesothelial signature in visceral tissue when compared with the subcutaneous fraction, our results were reproduced in ASCs, which were isolated by a gold-standard method accepted by both the International Society of Cell Therapy and the International Federation for Adipose Therapeutics and Science [62], and presented an expression profile unique in human ASC populations [63,64,65,66]. Moreover, our data are in line with a recent single-cell sequencing study, which detected a mesothelial-enriched subpopulation among VAT precursors [17]. Thus, we believe that there was no mesothelial contamination among the visceral precursors, and that our results present solid evidence in favor of the mesothelial origin of visceral fat precursors and enrichment in creeping fat. Since prospective lineage-tracking is not applicable to humans [67], future studies will need to combine single-cell transcriptomics with better sorting approaches for a more accurate functional characterization of human adipocyte precursor subpopulations. Along this line, several studies already identified distinct subtypes of human adipocyte precursors [12,14,16,22] and correlated the altered frequency of these with WAT fibrosis and the severity of insulin resistance and T2D [17,22,25]. In rodents, Chau et al. [21] proposed that *Wt1*-lineage adipocytes had fewer but larger lipid droplets per cell than adipocytes arising from *Wt1*-negative progenitors. More recently, other studies identified an adipose tissue mesothelial-like subpopulation [23] that showed decreased insulin-induced glucose uptake and triglyceride accumulation, and weakened *de novo* lipogenesis [15]. However, adipocyte precursors deriving from distinct lineages, and isolated from the same depot, exhibited indistinguishable responses to a high-fat diet, indicating that ontogenetic differences between adipose progenitors do not necessarily correspond to functional differences in this context [23]. Altogether, these findings suggest that the behavior of adipocyte lineage cells is not strictly determined by developmental history, putting emphasis on the recruitment of adult precursors to shape the properties of adipose tissue. We can also speculate that the physiological state of adipose tissue could favor the proliferation of specific mesenchymal or mesothelial progenitors to condition the tissue’s properties. Targeted ASCs within creeping fat in Crohn’s disease patients may be recommended to control the exacerbated immune response in these patients and may be critical to control the severity of Crohn’s disease. Additionally, the site of inflammation (either ileal or colonic) seems also to determine the properties and composition of mesenteric adipose tissue [68] and should be considered in future investigations.

## 4. Materials and Methods

### 4.1. Study Subjects

Subjects were recruited at the University Hospital Joan XXIII (Tarragona, Spain) and University Hospital Vall d’Hebrón (Barcelona, Spain). All participants gave their informed consent, and the study was reviewed and approved by the ethics and research committees of their respective hospitals (references CEIM 177/2018 approved on the 29th of November 2018 and CEIM 41p/2015 approved on the 31st of August 2015) in accordance with the tenets of the Helsinki Declaration. Donors were classified according to their body-mass index (BMI) as lean, obese, or morbid obese following World Health Organization criteria (lean BMI ≤ 25 kg/m^2^, obese BMI ≥ 30 ≤ 35 kg/m^2^, and morbid obese BMI ≥ 40 kg/m^2^). All subjects in the study were stratified according to age, gender, and BMI. The recruited subjects were not diabetic, nor had any underlying pathology on physical examination and tests other than those associated with an excess of weight or Crohn’s Disease. Anthropometric and biochemical variables from the cohort are detailed in Table 1**.** Regarding the Montreal classification of Crohn’s disease patients [69], with respect to the age onset, 10% of these were A1 (diagnosed under the age of 17), 42% were A2 (in the 17- to 40-year age group), and 16% were A3 (older than 40 years of age). Regarding the disease location, 42% of Crohn’s disease patients were L1 (ileal), 6% were L2 (colonic), and 19% were L3 (ileocolonic). As for the disease behavior, it was B2 (structuring) in 39% of patients and B3 (penetrating) in 29% of patients. Crohn’s disease patients were on immunomodulatory (32%), steroid (25%), and biological (25%) treatments.

As for the collection of samples, in the case of the lean and obese groups, VAT (omental tissue) and SAT were sampled during scheduled, non-acute surgical procedures, including laparoscopic surgery for hiatus hernia repair or cholecystectomies, whereas samples from patients with morbid obesity were obtained during bariatric surgery. Adipose tissue (VAT and SAT) was obtained from patients with Crohn’s disease undergoing surgical resection for symptomatic complications or failure of medical therapy. Specifically, VAT was sampled from the mesenteric creeping fat. To guarantee the purity of the adipose tissue explants, especially VAT, any tissue (e.g., mesothelial layer [70]) other than fat was completely removed before processing the biopsy.

Fat samples for proteomic studies were obtained from three lean males (46.67 ± 13.28 years of age and BMI 21.93 ± 0.98 kg/m^2^) during hiatal hernia surgery. Subjects were recruited at the Endocrinology and Nutrition Unit of the Hospital Clínico Virgen de la Victoria (Málaga, Spain). Samples were washed in physiological saline and immediately frozen in liquid nitrogen and maintained at −80 °C until analysis. Participants gave their informed written consent. The ethics and research committee of the Hospital Clínico Virgen de la Victoria approved the protocol. All samples are stored in a tissue biobank registered at the National Register of Biobanks (C.0003609).

### 4.2. Adipose Stem Cell Isolation and Cell Culture

Fat tissue was washed in phosphate buffered saline (PBS) and immediately frozen in liquid N_2_ with storage at −80 °C, or was used immediately for ASC isolation, as described [62]. Briefly, freshly isolated VAT was washed in PBS and treated with 2 mg/mL collagenase type I (Sigma-Aldrich, St. Louis, MO, USA) in PBS 1% bovine serum albumin at 37 °C for 1 h with gentle agitation. Digested samples were centrifuged at 300× *g* for 5 min to separate adipocytes from stromal cells. The cell pellet containing the stromal fraction was resuspended in Dulbecco’s modified Eagle’s medium high-glucose/Ham’s F12 medium (1:1), 10% fetal bovine serum, and 1% antibiotic/antimycotic solution (penicillin, streptomycin, and amphotericin). Cells were incubated at 37 °C in a humidified atmosphere containing 5% CO_2_/21% O_2_. The medium was replaced 24 h after seeding to remove non-adherent cells and was replenished every 2–3 days thereafter. All experiments were performed in cells at passages 3–7.

### 4.3. Immunophenotyping of ASCs

Flow cytometry analysis of cell marker expression confirmed that the isolated ASCs showed minimal functional and quantitative criteria as established by the International Society of Cell Therapy and the International Federation for Adipose Therapeutics and Science. Aliquots of ASCs (2 × 10^5^ cells) were incubated with a panel of primary antibodies (BD Pharmingen, San Diego, CA, USA) for 20 min at RT (protected from light), as previously described [6,7,66]. ASCs were positive for the expression of typical surface antigens, including CD90 (Thy-1 antigen; >96%), CD73 (ecto-5′-nucleotidase; >98%), CD105 (Endoglin; >82%), and CD36 (platelet glycoprotein 4; >43%), and negative for CD34 (hematopoietic progenitor cell antigen CD34; <0.1%), CD45 (leukocyte common antigen; <0.1%), CD14 (monocyte differentiation antigen; <2%), and CD31 (platelet endothelial cell adhesion molecule; <1%). In addition, we analyzed those related to antigen presentation such as HLA-DR (Human Leukocyte Antigen—DR isotype), CD86 (T-lymphocyte activation antigen CD86), and CD40 (Tumor necrosis factor receptor superfamily member 5).

### 4.4. Gene Expression Analysis

Fat tissue samples were homogenized using a conventional battery-operated, handheld homogenizer and solubilized in Tris Reagent (Sigma, St. Louis, MO, USA). Total RNA was isolated from adipose tissue and cells using the miRNeasy Mini kit (Qiagen Science, Hilden, Germany). RNA quantity was measured at 260 nm, and purity was assessed by the OD260/OD280 ratio. One microgram of RNA was transcribed to cDNA with random primers using the Reverse Transcription System (Applied Biosystems, Foster City, CA, USA). Quantitative gene expression was evaluated by real-time polymerase chain reaction (qPCR) on a 7900HT Fast Real-Time PCR System, using 384-well microfluidic TaqMan low-density array cards with an inventoried panel of gene expression assays from Applied Biosystems (Carlsbad, CA, USA). We performed an extensive literature search to assemble the mesothelial and mesenchymal expression profiles with well-established markers (Appendix A). Relative expression levels were calculated using the comparative Ct method and normalized to the expression of the housekeeping genes cyclophilin 1A (PPIA; reference gene for the tissue samples) and 18S (reference gene for ASCs samples).

### 4.5. Proteomics

Paired SAT and VAT samples (100 mg) from three lean male subjects were pooled and processed for proteomic analysis, as described [71]. Samples were pooled to reduce biological variance among individuals. LC-MS/MS analysis was carried out at the Proteomics Unit of the University of Córdoba (Central Services Research Support, Córdoba, Spain). Adipose tissue samples were analyzed using a Dionex Ultimate 3000 nano-liquid chromatography system (nano UHPLC; Thermo Fisher Scientific, Sunnyvale, CA, USA) in tandem with a linear quadrupole ion trap Orbitrap (LTQ Orbitrap XL) mass spectrometer equipped with a nanoelectrospray ion source (Thermo Fisher Scientific, Waltham, MA, USA). Acquired data were analyzed with Proteome Discoverer 2.1 software (Sequest HT algorithm; Thermo Fisher Scientific, Waltham, MA, USA), using a specific human database extracted from the non-redundant UNIPROT website (https://www.uniprot.org (accessed on 20 April 2021).). Peptide searches were performed with a maximum of one missed cleavage site. Carbamidomethylation of cysteines was set as a fixed modification, and methionine oxidation and lysine acetylation were set as variable modifications. Finally, the identified peptides were filtered using a 1% false-discovery rate calculated using a decoy database strategy.

### 4.6. Antigen Presentation Assay between ASCs and Autologous CD4^+^ T-lymphocytes

This assay required a coordinated extraction of visceral adipose tissue and isolation of ASCs with the collection of blood samples from the same patient. ASCs were kept for approximately three weeks in culture, and when they reached passage number 3, an additional appointment with the patient took place to collect the blood samples and isolate peripheral blood mononuclear cells (PBMCs) and CD4^+^ T-lymphocytes. The detailed antigen presentation protocol was as follows:

Day 1: ASCs activation. Confluent ASCs cultures were treated or not with Interferon gamma (IFNγ) (10 ng/mL) for 1 h before the addition or not of ovalbumin (OVA) (1 mg/mL) for 18 h [72,73].

Day 2: CD4^+^ T-cells isolation and co-culture. CD4^+^ T-cells were isolated from PBMCs, which were obtained from heparinized blood by gradient centrifugation on a Ficoll solution (density 1.077 g/mL, Biochrom, Germany) at 800× *g* for 15 min at room temperature. PBMCs’ count and viability was assessed by trypan blue dye exclusion, and cells were immediately used for CD4^+^ T-cells’ isolation through the autoMACS Pro Cell Separator using the human CD4^+^ T-cell Isolation Kit (Miltenyi Biotec, Bergisch Gladbach, Germany), according to the manufacture’s protocol. Autologous CD4^+^ T-cells were labeled with 2 μM CellTrace violet dye (CTV) (CellTrace kits; Invitrogen, Waltham, MA, USA) according to manufacturer’s protocol, after which were added to ASCs at a 1:4 ratio in RPMI medium.

Day 4: assay. T-cells’ proliferation was followed by flow cytometry by examining CTV intensity in viable CD4^+^ T-lymphocytes. ASCs were harvested and immunophenotyped, as previously described. Data were acquired on FACSAria™ III (BD Biosciences, Franklin Lakes, NJ, USA) and were analyzed using FACSDiva™ v8 (BD Biosciences, Franklin Lakes, NJ, USA) and FlowJo™ v10 softwares (Ashland, OR, USA).

### 4.7. Statistical Analysis

Statistical analysis was performed with the statistical computation system R, Version 2.6.2 [74]. The results were expressed as mean ± SEM of *n* biologically independent samples. Significance was tested by the unpaired Mann-Whitney U test with Benjamin-Hochberg adjustment. Linear models with empirical Bayes statistic (Limma) were used for data obtained from the National Center for Biotechnology Information Gene Expression Omnibus (GEO) database.

## 5. Conclusions

We found a mesothelial gene signature that distinguishes visceral adipose tissue and its precursors from subcutaneous adipose tissue. Furthermore, we demonstrated that the creeping fat precursor cells have mesothelial-enriched properties including an altered immunophenotypic profile and may act as antigen-presenting cells.

## Figures and Tables

**Figure 1 ijms-22-04292-f001:**
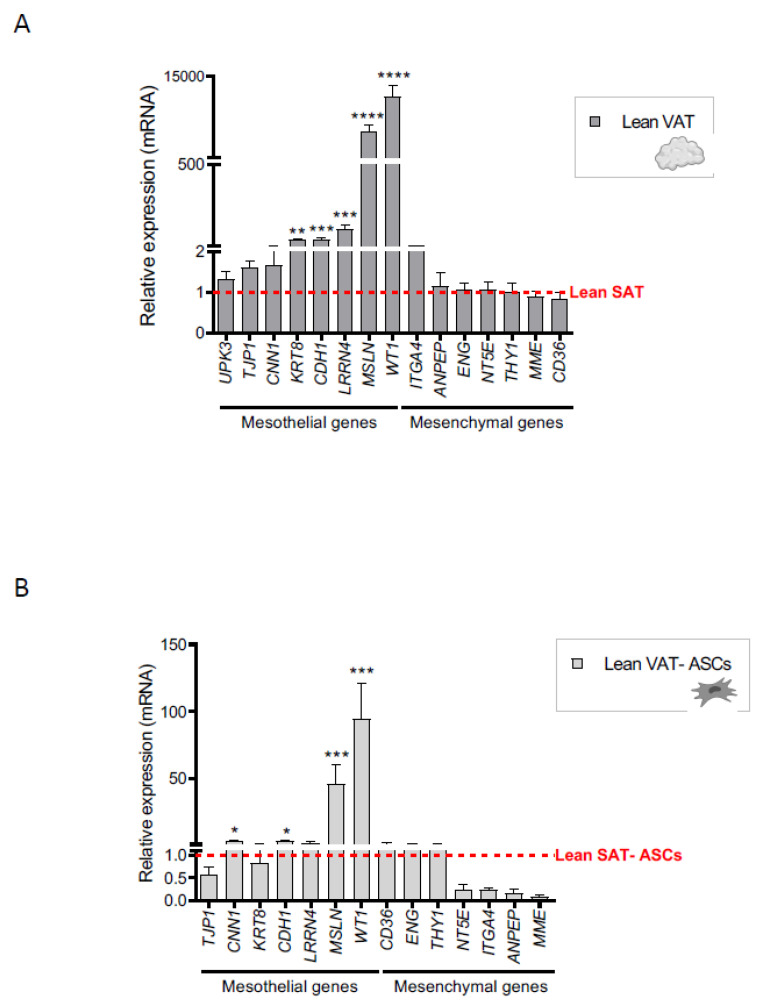
Mesothelial markers are elevated in lean VAT and in VAT-derived ASCs. qPCR analysis of mesothelial and mesenchymal marker expression in (**A**) Lean VAT versus lean SAT samples (VAT *n* = 10; SAT *n* = 12) and (**B**) VAT-derived ASCs versus SAT-derived ASCs (VAT-ASCs *n* = 8, except *n* = 14 for *LRRN4*, *MSLN*, *KRT8*, *CDH1*, *WT1*, *CDH2*, and *CA2*; SAT-ASCs *n* = 8). Mean lean SAT values were normalized to 1, and are represented as a red dotted line. *n* represents biologically independent samples. Abbreviations: *ANPEP*, alanyl aminopeptidase; ASCs, adipose-derived stem cells; *CNN1*, calponin 1; *CDH1*, cadherin 1; *ENG*, endoglin; *ITGA4*, integrin subunit alpha 4; *KRT8*, keratin 8; *LRRN4*, leucine rich repeat neuronal 4; *MME*, membrane metalloendopeptidase; *MSLN*, mesothelin; *NT5E*, 5’-nucleotidase ecto; SAT, subcutaneous adipose tissue; *TJP1*, tight junction protein 1; *UPK3*, uroplakin 3A; VAT, visceral adipose tissue; *WT1*, Wilms tumor protein. * *p* < 0.05; ** *p* < 0.01; *** *p* < 0.001; **** *p* < 0.0001 (Mann-Whitney U test).

**Figure 2 ijms-22-04292-f002:**
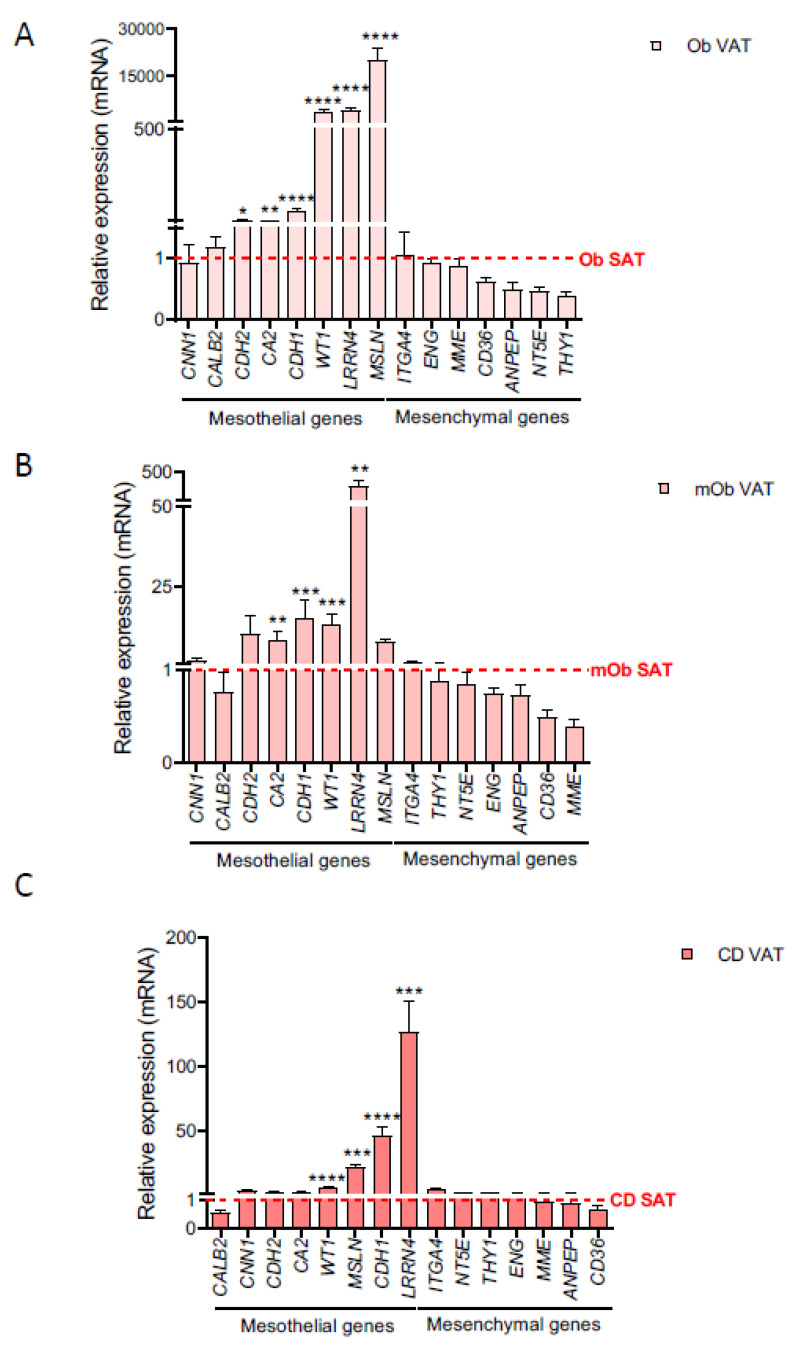
Mesothelial markers are upregulated in obese, morbid obese, and Crohn’s disease VAT with respect to SAT. qPCR analysis of mesothelial and mesenchymal marker gene expression in VAT versus SAT samples in (**A**) obese (ob) (ob VAT *n* = 10, except *n* = 14 for *LRRN4*, *MSLN*, *KRT8*, *CDH1*, *WT1*, *CDH2,* and *CA2*; ob SAT *n* = 10), (**B**) morbid obese (mob) (mob VAT *n* = 13 or 8, and mob SAT *n* = 8) and (**C**) Crohn’s disease (CD VAT *n* = 10, except *n* = 20 for *LRRN4*, *MSLN*, *KRT8*, *CDH1*, *WT1*, *CDH2*, and *CA2*; CD SAT *n* = 8). Mean SAT levels were normalized to 1, and are represented as a red dotted line. Abbreviations: *CA2*, carbonic anhydrase 2; *CALB2*, calretinin; CD, Crohn’s disease; ob, obese; mob, morbid obese; SAT, subcutaneous adipose tissue; VAT, visceral adipose tissue. Data are expressed as mean ± SEM; * *p* < 0.05; ** *p* < 0.01; *** *p* < 0.001; **** *p* < 0.0001 (Mann-Whitney U test).

**Figure 3 ijms-22-04292-f003:**
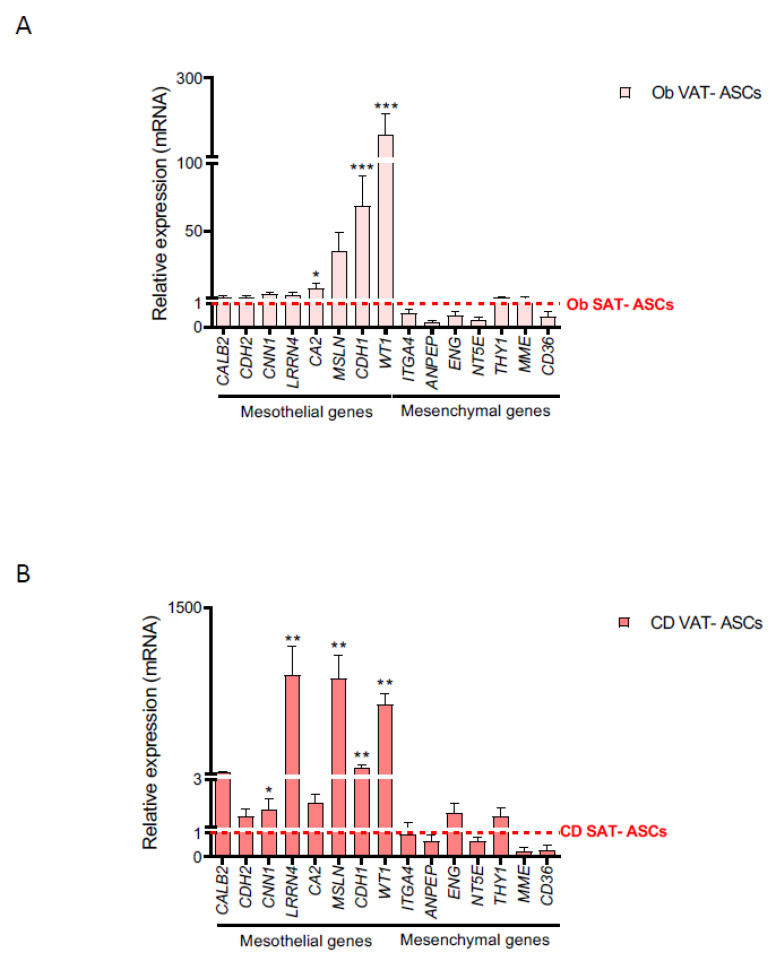
Mesothelial markers are upregulated in obese and Crohn’s disease VAT-derived ASCs with respect to SAT-derived ASCs. qPCR analysis of mesothelial and mesenchymal marker gene expression in VAT-derived ASCs versus SAT-derived ASCs in (**A**) obese (ob VAT *n* = 4, except *n* = 10 for *LRRN4*, *MSLN*, *KRT8*, *CDH1*, *WT1*, *CDH2*, and *CA2*; ob SAT *n* = 8), and (**B**) Crohn’s disease (CD VAT *n* = 13, except *n* = 28 for *LRRN4*, *MSLN*, *KRT8*, *CDH1*, *WT1*, *CDH2*, and *CA2*; CD SAT *n* = 4). Mean SAT values were normalized to 1, and are represented as a red dotted line. Green-red scale is used where green indicates higher expression and red lower expression. Abbreviations: CD, Crohn’s disease; SAT, subcutaneous adipose tissue; VAT, visceral adipose tissue. Data are expressed as mean ± SEM; * *p* < 0.05; ** *p* < 0.01; *** *p* < 0.001 (Mann-Whitney U test).

**Figure 4 ijms-22-04292-f004:**
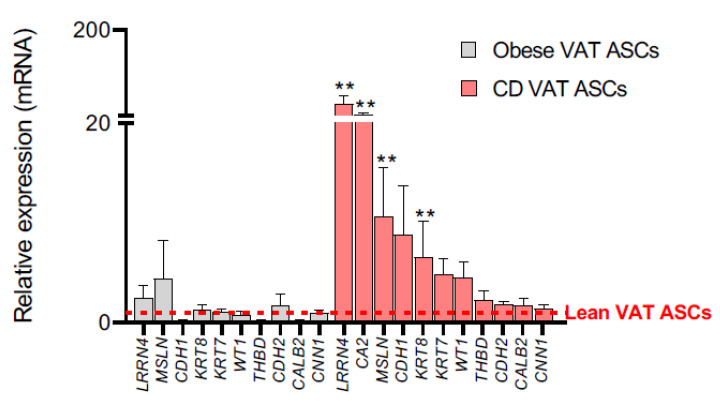
Mesothelial marker expression in VAT-derived ASCs from patients with Crohn’s disease is upregulated with respect to equivalent lean progenitors. qPCR analysis of mesothelial marker expression in obese and Crohn’s disease VAT-derived ASCs versus equivalent lean-derived cells. Mean lean VAT-derived ASC values were normalized to 1, and are represented as a red dotted line. Abbreviations: ASCs, adipose-derived stem cells; CD, Crohn’s disease; ob, obese; *THBD*, thrombomodulin; VAT, visceral adipose tissue. Data are expressed as mean ± SEM; ** *p* <0.01 (Mann-Whitney U test with Benjamin-Hochberg adjustment).

**Figure 5 ijms-22-04292-f005:**
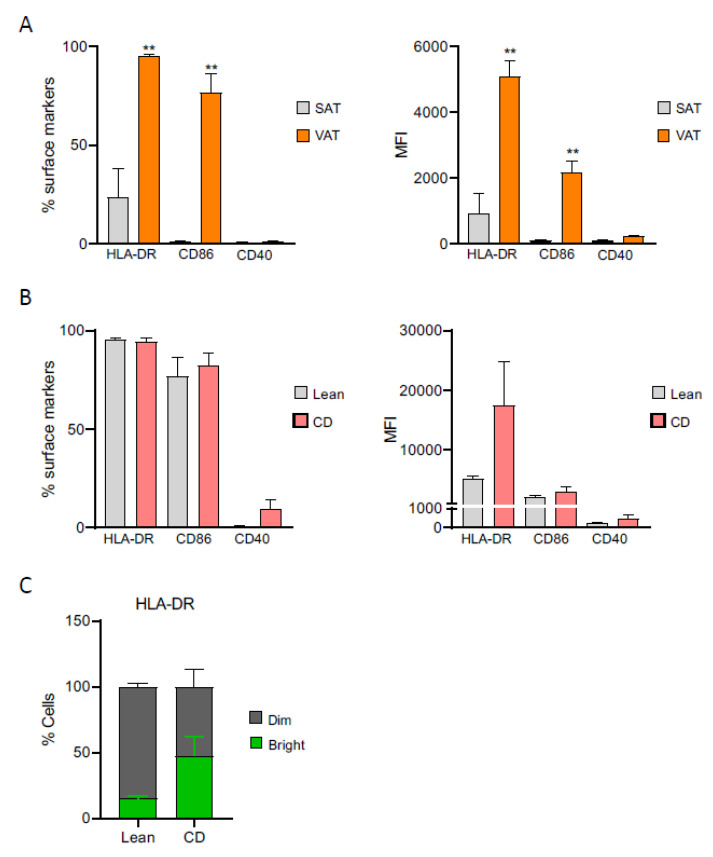
Creeping fat-derived ASCs have an altered immunophenotypic profile with respect to their lean counterparts. ASCs were stained with the panel of antibodies described and analyzed by flow cytometry using a FACSAria III (BD Biosciences, Franklin Lakes, NJ, USA) equipped with three lasers operating at 405, 488, and 633 nm. Surface markers were detected using phycoerythrin (PE) (585 nm band-pass filter), PE-Cy7 (780 nm band-pass filter), allophycocyanin (APC) (660 nm band-pass filter), Fluorescein isothiocyanate (FITC) (530 nm band-pass filter), Peridinin-Chlorophyll-Protein (PerCP-cy5.5) (695 nm band-pass filter), and BD Horizon Brilliant Violet 421 channels (450 band-pass filter). Gating was based on forward-scatter and side-scatter dot plots. A side-scatter versus Ho342 dot plot was used to discriminate nucleated cells from debris. Analysis of antigen-presenting molecules compared the (**A**) percentage of surface expression and mean fluorescence intensity (MFI) between subcutaneous (SAT)-derived ASCs versus visceral (VAT)-derived ASCs from lean patient and the (**B**) percentage of surface expression and MFI between lean-derived ASCs versus Crohn’s disease-derived ASCs, from visceral and mesenteric adipose tissue, respectively. (**C)** Percentage of HLA-DR subpopulations HLA-DR ^dim^ and HLA-DR ^bright^ in lean-derived and Crohn’s Disease-derived ASCs. Abbreviations: CD, Crohn’s disease; ob, obese; HLA-DR, human leucocyte antigen—DR isotype; VAT, visceral adipose tissue; SAT, subcutaneous adipose tissue. Data are expressed as the mean ± SEM; ** *p* < 0.01 (Mann-Whitney U test with Benjamin-Hochberg adjustment).

**Figure 6 ijms-22-04292-f006:**
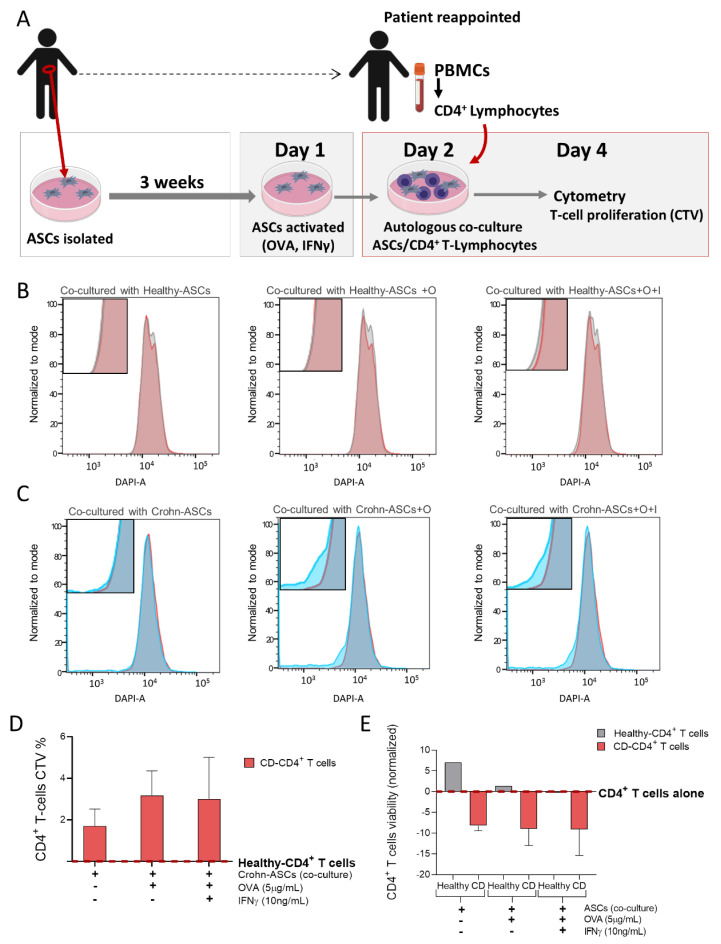
Creeping fat-derived ASCs act as antigen-presenting cells. (**A**) Schematic representation of the antigen-presentation assay. CD4^+^ T-lymphocytes were co-cultured with autologous healthy- and Crohn’s disease-derived ASCs, from visceral and mesenteric adipose tissue, respectively. T-cell proliferation was measured by CellTrace violet dye (CTV) intensity after two days of co-culture with untreated ASCs, and ASCs pulsed with ovalbumin (OVA), in the presence or absence of IFNγ. (**B**,**C**) Representative histograms for healthy- and Crohn’s Disease-derived CD4^+^ T-lymphocytes alone (grey and dark blue, respectively) and CD4^+^ T-lymphocytes co-cultured with healthy- and Crohn’s Disease-derived ASCs (red and lighter-blue, respectively). (**D**) Crohn’s Disease-derived CD4^+^ T-lymphocytes’ (*n* = 3) proliferation relative to healthy control. (**E**) Healthy- and Crohn’s Disease-derived CD4^+^ T-lymphocytes’ viability relative lo lymphocytes alone. Abbreviations: CD, Crohn’s disease; INFγ and I, Interferon gamma; O and OVA, ovalbumin; PBMCs, peripheral blood mononuclear cells; VAT, visceral adipose tissue; SAT, subcutaneous adipose tissue. Data are expressed as the mean ± SEM.

**Table 1 ijms-22-04292-t001:** Anthropometric and biochemical variables of subjects.

Variables	Lean	Obese	Morbid Obese	CD
*n*	33	14	14	31
Sex (male/female)	13/20	5/9	4/10	17/15
Age (years)	48.12 ± 14.48	50.32 ± 11.08	55.28 ±12.86	39.55 ± 13.85 *
BMI (kg/m^2^)	23.12 ± 2.10	31.87 ± 1.76 †	37.73 ± 2.86 †	23.15 ± 4.68
Glucose (mg/dL)	89.22 ± 15.32	97.93 ± 16.69	97.14 ± 14.63 *	87.60 ± 32.00
Cholesterol (mg/dL)	178.99 ± 33.01	211.86 ± 34.37	187.69 ± 35.80	134.02 ± 33.71 †
HDLc (mg/dL)	54.65 ± 14.20	45.17 ± 12.81	45.58 ± 12.30	46.00 ± 23.94 *
Triglycerides (mg/dL)	116.04 ± 85.34	149.84 ± 59.90 *	183.08 ± 123.69 *	125.33 ± 73.12

Abbreviations: BMI, body mass index; CD, Crohn’s disease; HDLc, high-density lipoprotein cholesterol. Differences were analyzed by the unpaired *t*-test. Data are presented as mean ± SD. * < 0.05, † < 0.0001 with respect to lean.

## Data Availability

Original imaging data have been deposited to Mendeley Data and are available http://dx.doi.org/10.17632/w6cnxgyrph.1 (accessed on 20 April 2021).

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
