# Peer review of "Crohn’s Disease Increases the Mesothelial Properties of Adipocyte Progenitors in the Creeping Fat"

_ijms, 2021, doi:10.3390/ijms22084292_

Round 1

Reviewer 1 Report

The manuscript entitled “Crohn’s disease increases the mesothelial properties of adipocyte progenitors in creeping fat” focuses on the mesothelial/mesenchymal properties of human adipose tissue and its precursors in Crohn’s disease and obesity.

The manuscript is well written, it has a complete experiment design, and the discussion is detailed and correct. The results obtained are clear. However, I have some comments:

-Have you quantified expression of genes related to inflammation?

-It would be nice to corroborate gene expression differences and their predictive association with metabolic phenotypes

Reviewer 2 Report

The present report from Madeira et al. characterizes creeping fat in Crohn’s disease patients. According to the findings described, creeping fat originates from mesothelial progenitors, according to its genetic and proteomic profile, and exhibits characteristic immune properties.

Several issues must be addressed by the authors, as indicated below.

Numerous abbreviations are used along the manuscript. In all cases, abbreviations must be spell out upon their first appearance.

Lean VAT-ASCs were characterized by an increased expression of mesothelial genes, when compared with SAT-ASCs. However, a reduction in the expression of several mesenchymal genes is also noted. This should be indicated in the text and the corresponding figure (Fig. 1B). The same applies for the expression of mesenchymal genes in Crohn’s disease patients vs. obese/morbid obese patients. This should be indicated.

When assessing the immunophenotypic profile of creeping fat-derived ASCs, some of the differences stated between lean and CD samples are not clear. For instance, according to the text, there is a difference in HLA-DRdim and HLA-DRbright subpopulations, however the corresponding figure (Fig, 5C) does not indicate any difference. These discrepancies should be reviewed and corrected adequately.

In all cases, the number of subjects/samples under study must be clearly stated. In some cases, and given the low n (in particular: n=2 for obese, morbid obese and Crohn’s disease SAT and SAT-derived ASCs simples), the individual values obtained, rather than a mean±SEM, must be provided.

Along the results section, discussion-related statements must be avoided. These considerations must be moved to the discussion section whenever necessary.

Fig. 7 is not clear, does not improve the understanding of the data and should be removed.

Table 1 (Anthropometric and biochemical data of subjects) is not included in the report.

For gene expression analysis, two housekeeping genes, namely cyclophilin 1A and 18S, were used. How they were used? Which gene was selected as reference for the different markers assessed?.

The authors must be more specific with their final conclusion, taking into account the results of their study. Current conclusion is too broad and speculative.

Specific comments:

Page 5, line 11: Spell out “LC-MS/MS”

Page 5, lines 19-23: Avoid discussion-related statements. These considerations should be moved to the discussion section.

Page 7, lines 19-23: Avoid discussion-related statements. These statements should be moved to the discussion section.

Page 12, lines 1-2: Move to discussion section.

Page 13, lines 1-3: Move to discussion section.

Page 14, lines 18-20: Avoid discussion-related statements. These statements should be moved to the discussion section.

Fig. 6A: Relates to the methods.

Page 23, lines 1-2: Provide reference number for ethics approval. This must also be included in the “Institutional Review Board Statement”.

Reviewer 3 Report

The manuscript about adipocyte stem cells in Crohn patients investigates the mesothelial characteristics of these cells in comparison to adipocyte precursors in obese patients. Interestingly, the phenotype of these cells differs considerably. 

Moreover, ASC derived from Crohn patients express more HLA-DR than control counterparts. With this characteristic, they can activate CD4 cells and modulate adaptative immunity.

The paper itself is clear and the results are well presented and supported by the discussion. Only some minor spell checks are required throughout the text.  It would also be helpful to increase the quality of figure 6 as some text is difficult to read.

Lastly, it would be interesting to know if ASC derived from Crohn patients can secrete proinflammatory cytokines (maybe more than their healthy control) as an additional way to activate CD4 cells. It is worth verifying the higher expression of TNF and IL-6 seen in this review https://doi.org/10.3390/biom9120780
